# The Role of the Gut Microbiota in the Effects of Early-Life Stress and Dietary Fatty Acids on Later-Life Central and Metabolic Outcomes in Mice

Kitty Reemst,[a] Sebastian Tims,[b] Kit-Yi Yam,[a] Mona Mischke,[b] Jan Knol,[b] (ORCID) Stanley Brul,[a] Lidewij Schipper,[b] (ORCID) Aniko Korosi[a]

[a]Swammerdam Institute for Life Sciences, Universiteit van Amsterdam, Amsterdam, the Netherlands
[b]Danone Nutricia Research, Utrecht, the Netherlands

**ABSTRACT** Early-life stress (ELS) leads to increased vulnerability for mental and metabolic disorders. We have previously shown that a low dietary $\omega$-6/$\omega$-3 polyunsaturated fatty acid (PUFA) ratio protects against ELS-induced cognitive impairments. Due to the importance of the gut microbiota as a determinant of long-term health, we here study the impact of ELS and dietary PUFAs on the gut microbiota and how this relates to the previously described cognitive, metabolic, and fatty acid profiles. Male mice were exposed to ELS via the limited bedding and nesting paradigm (postnatal day (P)2 to P9 and to an early diet (P2 to P42) with an either high (15) or low (1) $\omega$-6 linoleic acid to $\omega$-3 alpha-linolenic acid ratio. 16S rRNA was sequenced and analyzed from fecal samples at P21, P42, and P180. Age impacted $\alpha$- and $\beta$-diversity. ELS and diet together predicted variance in microbiota composition and affected the relative abundance of bacterial groups at several taxonomic levels in the short and long term. For example, age increased the abundance of the phyla *Bacteroidetes*, while it decreased *Actinobacteria* and *Verrucomicrobia*; ELS reduced the genera RC9 gut group and *Rikenella*, and the low $\omega$-6/$\omega$-3 diet reduced the abundance of the *Firmicutes Erysipelotrichia*. At P42, species abundance correlated with body fat mass and circulating leptin (e.g., *Bacteroidetes* and *Proteobacteria* taxa) and fatty acid profiles (e.g., *Firmicutes* taxa). This study gives novel insights into the impact of age, ELS, and dietary PUFAs on microbiota composition, providing potential targets for noninvasive (nutritional) modulation of ELS-induced deficits.

**IMPORTANCE** Early-life stress (ELS) leads to increased vulnerability to develop mental and metabolic disorders; however, the biological mechanisms leading to such programming are not fully clear. Increased attention has been given to the importance of the gut microbiota as a determinant of long-term health and as a potential target for noninvasive nutritional strategies to protect against the negative impact of ELS. Here, we give novel insights into the complex interaction between ELS, early dietary $\omega$-3 availability, and the gut microbiota across ages and provide new potential targets for (nutritional) modulation of the long-term effects of the early-life environment via the microbiota.

**KEYWORDS** early-life stress, diet, interventions, polyunsaturated fatty acids, microbiota, microbiome, gut-brain axis, polyunsaturated fatty acids

There is ample clinical and preclinical evidence that early-life stress (ELS) is associated with increased vulnerability to mental and metabolic health problems such as depression and inflammatory bowel disease (1–4). We and others have shown in recent years that chronic ELS induced in rodent models via the limited bedding and nesting material (LBN) paradigm (5, 6) leads to impaired cognitive functions and an altered metabolic profile (7, 8). Moreover, we demonstrated that early postnatal exposure to a diet with a low $\omega$-6 to $\omega$-3 polyunsaturated fatty acid (PUFA) ratio was able to protect

**Ad Hoc Peer Reviewer** Beiwen Zheng, the First Affiliated Hospital, College of Medicine, Zhejiang University

Address correspondence to Aniko Korosi, A.korosi@uva.nl.

The authors declare a conflict of interest. Authors ST, MM, JK and LS are employed by Danone Nutricia Research.

against the ELS-induced cognitive deficits without affecting the metabolic alterations (9). Currently, the exact underlying mechanisms for the effects of ELS and the beneficial effect of the diet are not fully understood and may be multifactorial. In this paper we address the effects of ELS and the dietary $\omega$-6/$\omega$-3 PUFA ratio on the fecal microbiota and if and how these relate to the effects of ELS and early postnatal diet on both the brain and metabolism across different ages that we reported earlier (9).

In recent years, there has been an increasing interest in how the gut microbiome might impact our health (10, 11). Particular attention has been devoted to the cross talk between the gut microbiota and the brain, known as the microbiota-gut-brain (MGB) axis, an integrated communication system including neural, hormonal, and immunological signaling pathways through which the gut microbiota can influence brain development and function and vice versa (12, 13). Increasing evidence supports the intriguing hypothesis that the microbiota can influence brain functions, that dysbiosis might contribute to changes in behavior (e.g., social behavior [14]) and the development and etiology of brain disorders (e.g., depression [15–18]), and that targeting the microbiota is effective in modulating brain function (e.g., cognitive functions [19]). Similarly, the gut microbiota is also thought to impact greatly on the immune system and metabolic health and has been associated with various risk factors of obesity and metabolic syndrome (20).

Several elements are emerging to be key in modulating the microbiome composition, including developmental life stages, stress, and diet (13, 21, 22). In fact, the development of the microbiome coincides with crucial (neuro)developmental periods. While little is known of the exact developmental trajectory of the microbiome in mice, we know from human literature that the intestinal microbiome starts to develop during and shortly after birth, during which time the brain is also going through immense developmental changes (23). Preclinical evidence shows that various early postnatal stress paradigms, in different species, impact the gut microbiota (24). For example, maternal separation (MS) has been shown to increase intestinal permeability in rats (4, 25) and affects the composition of the gut microbiota of infant rhesus monkeys directly after separation (26) and of 7-week-old rats (27). Such microbial composition changes may be instrumental for establishment of some of the MS-induced anxiety-related alterations, as germfree mice were not affected by MS to the same extent as colonized mice (28). Also, chronic ELS induced via the limited bedding and nesting material (LBN) paradigm (6) in male rats led to changes in microbiota composition and increased intestinal permeability at weaning age (29). Thus, the early-life adversity-induced dysbiosis could possibly contribute to later-life mental and metabolic health (10, 23, 24, 30, 31).

Next to development and exposure to early adversity, diet, and more specifically, dietary PUFA composition, has also been shown to modulate the composition of the gut microbiota at different stages of life (32, 33). For example, an 8-week supplementation with $\omega$-3 long-chain PUFAs (LCPUFAs), including docosahexaenoic acid (DHA) and eicosapentaenoic acid (EPA), in middle-aged healthy individuals led to multiple changes in bacterial taxa, including an increased abundance of genera involved in butyrate production, which have been suggested to be important for mental health (15, 34). The abundance of dietary $\omega$-3 and $\omega$-6 PUFAs during early life phases is also highly relevant, as these are key factors for proper development and function of the brain (35) and can influence the microbiome (33).

In the past century, there has been a marked change in the consumption of $\omega$-6 and $\omega$-3 PUFAs, with a high intake especially of $\omega$-6 linoleic acid (LA) in western societies, resulting in a high $\omega$-6/$\omega$-3 ratio (36). Given the relevance of dietary $\omega$-6/$\omega$-3 for brain development and function (35), this shift is thought to increase today's prevalence of psychopathology and chronic disease (37) and possibly also contributes to gut dysbiosis, thereby impacting the MGB axis (32). Therefore, dietary fatty acids have been explored as a possible strategy to modulate (stress-induced) behavioral changes and cognitive functioning (9, 38–41). In particular, the possible protective actions of $\omega$-3 PUFA during different life stages on the early-life stress-induced effects have been

explored (9, 40, 42). Pusceddu and colleagues demonstrated that long-term exposure to a diet with a low $\omega$-6/$\omega$-3 ratio (i.e., between 5 and 17 weeks of age by supplementation with $\omega$-3 LCPUFAs, including DHA) was beneficial for anxiety and cognition in nonstressed female rats and could restore part of the disturbed gut-microbiota composition of MS female rats, which was associated with the attenuation of the cortisol response to an acute stressor (40, 42). While this consisted of a lifelong intervention starting at 5 weeks of age, we have previously shown that a relatively short dietary intervention with a low-$\omega$-6/$\omega$-3 diet starting in the early postnatal period (i.e., from postnatal days 2 to 42) is able to restore the effects of ELS (via LBN) exposure on brain fatty acid (FA) composition early in life and on cognitive functions and brain plasticity in adulthood, without modulating the ELS-induced alterations in body fat mass and circulating leptin in mice (9).

Here, we study the effects of ELS, using the LBN paradigm in mice (postnatal day 2 (P2) and P9, an early dietary intervention with a low $\omega$-6 linoleic acid (LA) to $\omega$-3 alpha-linolenic acid (ALA) ratio (P2 and P42) and their interaction on the short-term (at P42) and long-term (at P180, after exposure to regular diet from P42 onward) impact on the gut microbiota composition and if and how these changes relate to the earlier reported central and metabolic ELS-induced profiles described in the same cohort of mice (9).

## RESULTS

**Age impacts $\alpha$- and $\beta$-diversity, and ELS and the dietary $\omega$-6/$\omega$-3 ratio affect $\beta$-diversity.** $\alpha$-diversity is the distribution of taxon abundances in a given sample into a single number that depends on both species richness and evenness and was tested by Chao1, Shannon, and phylogenetic diversity (PD). For all four experimental groups, the lowest $\alpha$-diversity within samples was observed at P21 and increased with age (generalized linear mixed model [GLMM], time point $P < 0.0001$ for all three measurements of $\alpha$-diversity) (Fig. 1B to D). No differences were detected in phylogenetic $\alpha$-diversity between the four experimental groups at any of the time points for any of the $\alpha$-diversity measures.

Our sample size at P21 was relatively low ($n = 3$ to 5 per group), and even though our methodology was able to pick up age-related changes in $\alpha$-diversity for further outcome measurements, we only analyzed the P42 and P180 time points.

**$\beta$-diversity.** Where $\alpha$-diversity focuses on community variation within a community (sample), $\beta$-diversity quantifies (dis-)similarities in microbiota composition between samples. Assessment of $\beta$-diversity at the genus level by permutational multivariate analysis of variance (PERMANOVA) showed a significant age effect on microbiota composition ($P < 0.0001$) with clustering of the four experimental groups at P42 (condition-diet interaction $P = 0.0064$) and P180 (condition-diet interaction $P = 0.0006$) (Fig. 1E). When performing distance-based redundancy analysis (db-RDA) per time point with litter correction, distinct clustering of the experimental groups was observed at P42 and P180 (Fig. 1F and H). At P42, the condition-diet interaction explained 12.8% of the total variation (with 10.6% in the first two db-RDA axes; Fig. 1F; significant with analysis of variance [ANOVA]-like permutation test for RDA, $P = 0.018$). For P180, the condition-diet interaction explained 13.9% of the total variation (with 11.9% in the first two db-RDA axes; see Fig. 1G; significant with ANOVA-like permutation test for RDA, $P = 0.003$).

In summary, $\alpha$-diversity was increased by age when looking at both species richness and evenness, which were not further affected by ELS or diet. Age also impacted phylogenetic $\beta$-diversity, and the interaction of the ELS and diet affected $\beta$-diversity, both at P42 and at P180.

**Fecal microbiota composition is affected by age, early-life stress, and the $\omega$-6/$\omega$-3 PUFA ratio of an early diet.** Analysis of relative abundances at the phylum, class, family, and genus levels shows that the fecal microbiota composition differed significantly for several bacterial taxa between the two time points (Fig. 1H) and between the experimental groups at both P42 (Fig. 2) and P180 (Fig. 3). All statistical differences are included in Table 1, and additional descriptive information on all measured

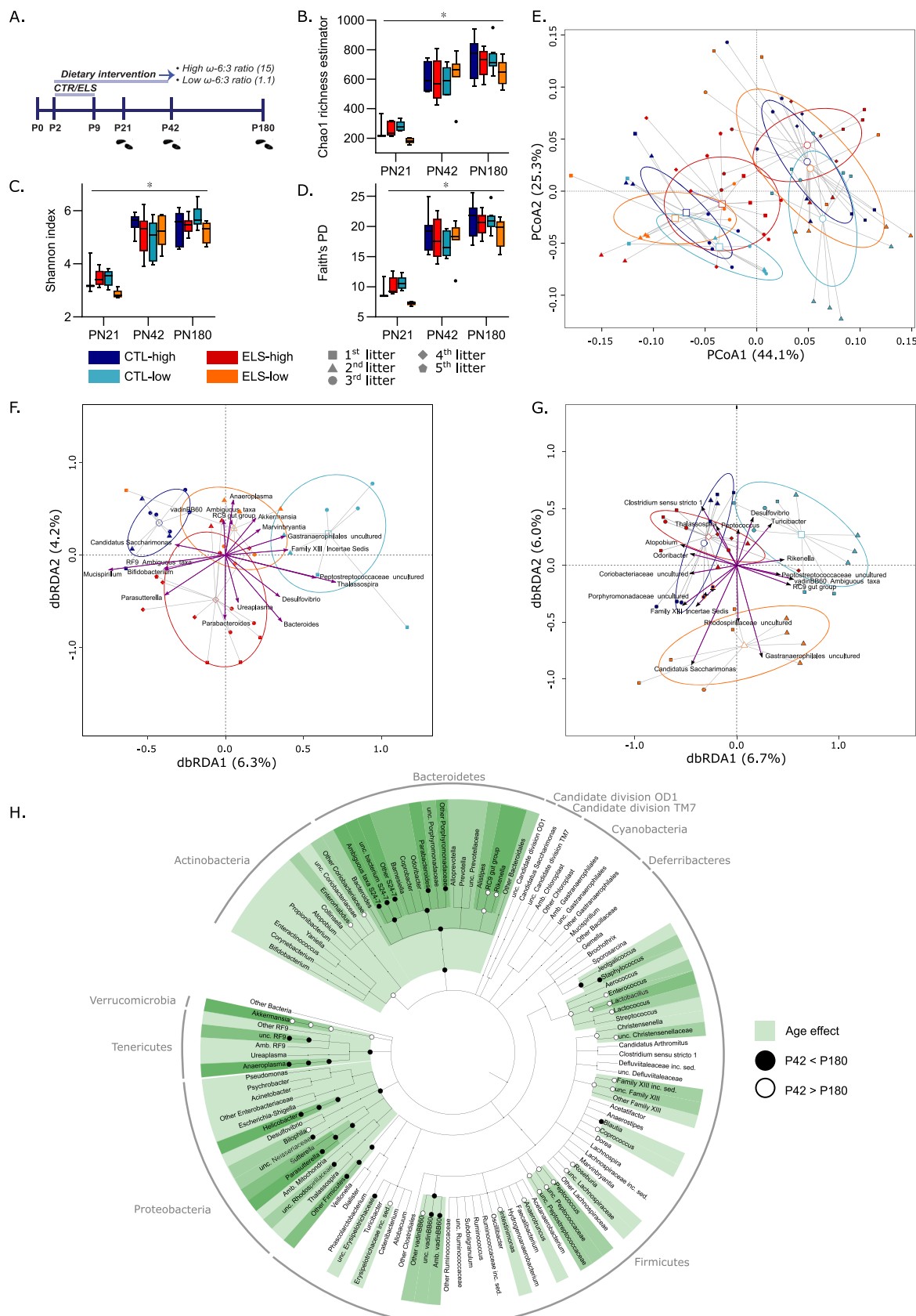

**FIG 1** Age impacts $\alpha$- and $\beta$-diversity and ELS and dietary $\omega$-6/$\omega$-3 ratio affect $\beta$-diversity, dependent on each other. (A) Experimental timeline. (B to D) Chao1 (A), Shannon (B), and phylogenetic diversity (C) plots displaying an increase in $\alpha$-diversity with age (GLMM at

bacterial species stratified per taxonomic level, age, and experimental group are included in Table S4 in the supplemental material. Analysis at the phylum level indicated that for both ages, the fecal microbiota was dominated by three major phyla, *Bacteroidetes*, *Firmicutes*, and *Verrucomicrobia*, but *Proteobacteria*, *Deferribacteres*, and *Actinobacteria* were also present (Fig. S1A). Other phyla detected in low abundance (<3%) were candidate division TM7 (2.1%), *Cyanobacteria* (0.67%), and *Tenericutes* (1.53%) (not depicted in Fig. S1A). An overview of the 20 most abundant genera for both ages are depicted in Fig. S1B.

Many changes were observed in the fecal microbiota composition of mice between P42 and P180 at all analyzed taxonomic levels (phylum, class, order, family, genus). At the phylum level, the abundance of *Bacteroidetes* increased with age, while *Actinobacteria* and *Verrucomicrobia* were found in lower abundances in P180 samples. At the genus level, among many others, *Parasutterella* and VadinBB60 increased and the RC9 gut group and *Bilophila* decreased with age. All age-mediated changes and statistical values are described in Fig. 1H and Table 2.

The main effects for ELS and the early dietary $\omega$-6/$\omega$-3 ratio on the relative abundance were detected for bacterial groups at P42 and P180 (Fig. 2; Fig. 3; Table 1). At P42, ELS exposure decreased the abundance of *Coprococcus*, and the low-$\omega$-6/$\omega$-3 diet reduced the class, order, and family *Erysipelotrichia*, *Erysipelotrichales*, and *Erysipelotrichaceae* belonging to *Firmicutes* (Fig. 2). At P180, the low-$\omega$-6/$\omega$-3 diet long-lastingly reduced the genus *Coriobacteriaceae uncultured*. ELS reduced the relative abundance of the genera RC9 gut group and *Rikenella*, both part of the *Rikenellaceae* family in adulthood at P180 (Fig. 3).

At both ages, most significant changes in the relative abundance of the microbiota were dependent on both ELS and dietary $\omega$-6/$\omega$-3 ratio (Table 1). At P42 (Fig. 2), interaction effects were found between ELS exposure and diet for the phylum *Cyanobacteria* and its class and order *Melainabacteria* and *Gastranaerophilales*, for which the low-$\omega$-6/$\omega$-3 diet significantly increased their abundance in specifically control (CTL) animals, while in ELS animals no differences were present dependent on the early diet. This same pattern was found for several *Clostridia* members; *Clostridiales* family XIII, an unassigned *Clostridiales* taxon, *incertae sedis*, and an uncultured family XIII taxon. Next, interaction effects between ELS and diet were detected for the class, order, family, and genus *Erysipelotrichia*, *Erysipelotrichales*, and *Erysipelotrichaceae*, and *Allobaculum*; the low-$\omega$-6/$\omega$-3 diet reduced its abundance in specifically CTL animals, while for ELS animals this reduction was not significant. Lastly, an interaction effect was found for the *Bacteroidetes* genus *Odoribacter*, for which ELS reduced its abundance in animals fed a high-$\omega$-6/$\omega$-3 diet but not in animals fed the low-$\omega$-6/$\omega$-3 diet.

At P180 (Fig. 3), an interaction between ELS and diet was found for the genus *Bifidobacterium*; its relative abundance was significantly higher in ELS-exposed animals fed the high-$\omega$-6/$\omega$-3 diet than in CTL and ELS-exposed animals fed the low-$\omega$-6/$\omega$-3 diet. For the bacterial group *Coriobacteriaceae uncultured*, except for the reduction by the low-$\omega$-6/$\omega$-3 diet for both CTL and ELS-exposed animals as described above, an interaction between ELS exposure and diet was found. Next, an interaction effect was found for three members of the *Rikenellaceae* family. ELS exposure increased the abundance of *Alistipes*, specifically in animals fed the high-$\omega$-6/$\omega$-3 diet. For the RC9 gut group and *Rikenella*, the ELS-induced reduction (main effect ELS as described above), was only significant in animals fed the low-$\omega$-6/$\omega$-3 diet. For the *Firmicutes* VadinBB60 ambiguous taxa and *Turicibacter*, the low-$\omega$-6/$\omega$-3 diet increased its abundance in CTL animals. Lastly, ELS exposure decreased the relative abundance of *Bilophila* only in ani-

**FIG 1** Legend (Continued)

sequencing depth of 11,535; $P < 0.0001$;). (E) $\beta$-diversity at the genus level analyzed by PERMANOVA showing effect of age ($P < 0.0001$) with clustering of the four experimental groups at P42 (condition-diet interaction $P = 0.0064$) and P180 (condition-diet interaction $P = 0.0066$). (F and G) db-RDA of $\beta$-diversity aggregated at the genus level for both ages separately. The 10 genera explaining most variation in the principal-component analysis (PCA) and db-RDA were visualized; (F) db-RDA at P42, ANOVA-like permutation test for RDA ($P = 0.018$); (G) db-RDA at P180, ANOVA-like permutation test for RDA ($P = 0.003$). (H) Cladogram showing significant age-mediated changes in relative abundance of bacterial species at several taxonomic levels. GLMM, general linear mixed model; db-RDA, distance-based redundancy analysis; ANOVA, (analysis of variance).

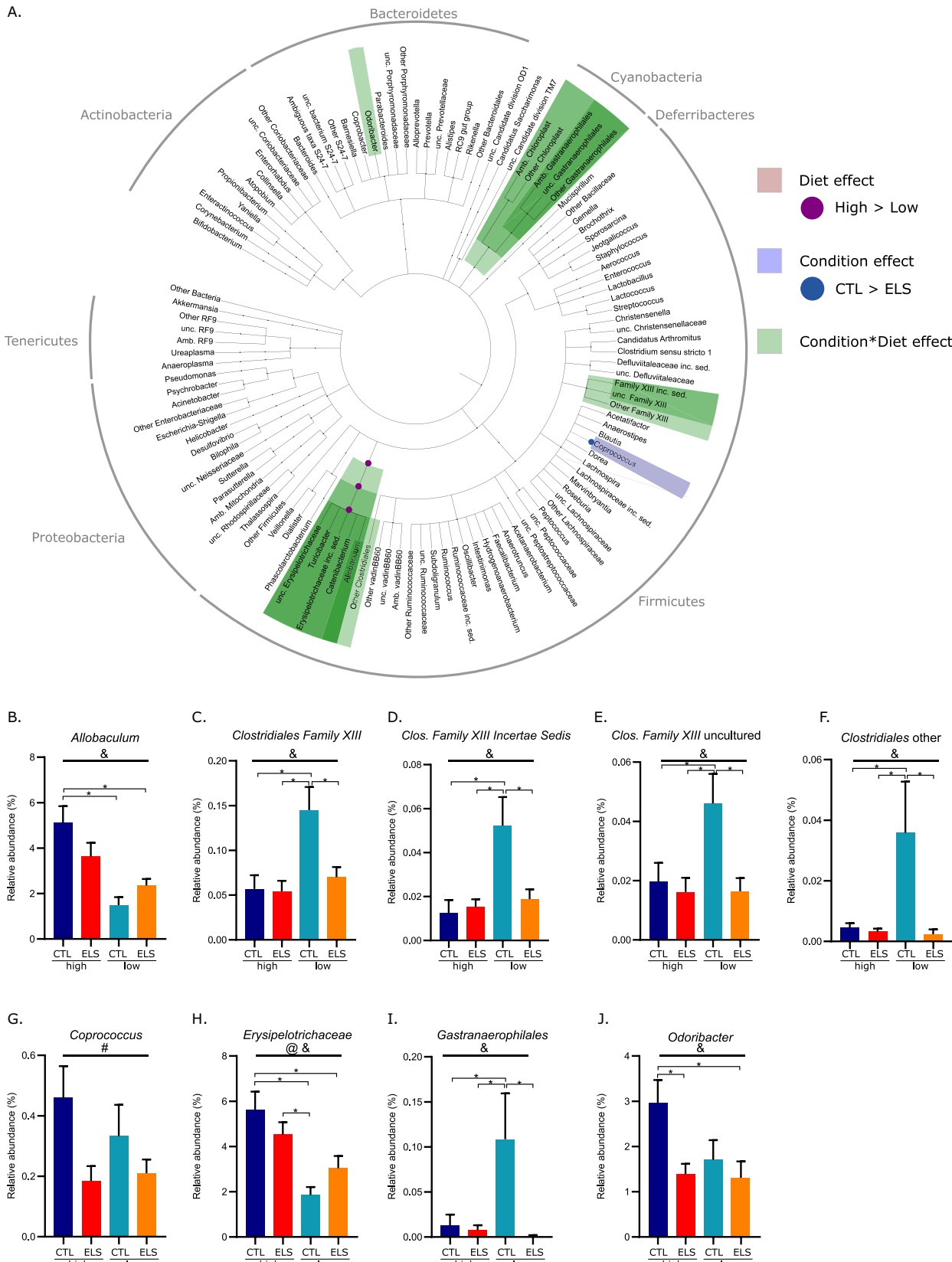

**FIG 2** Early-life stress and early dietary $\omega$-6/$\omega$-3 ratio affect the microbiota composition at P42 in interaction with each other. (A) Cladogram showing significant condition- and diet-mediated changes in the relative abundance of bacterial taxa at several taxonomic levels at P42. (B to J) Bar graphs of detected interaction effects (condition-diet) for bacterial taxa at P42 (GLMM $P < 0.05$ and $q < 0.1$). @, main effect of diet; &, interaction of condition-diet; ^, significant difference with Tukey *post hoc* test ($P < 0.05$); GLMM, general linear mixed model.

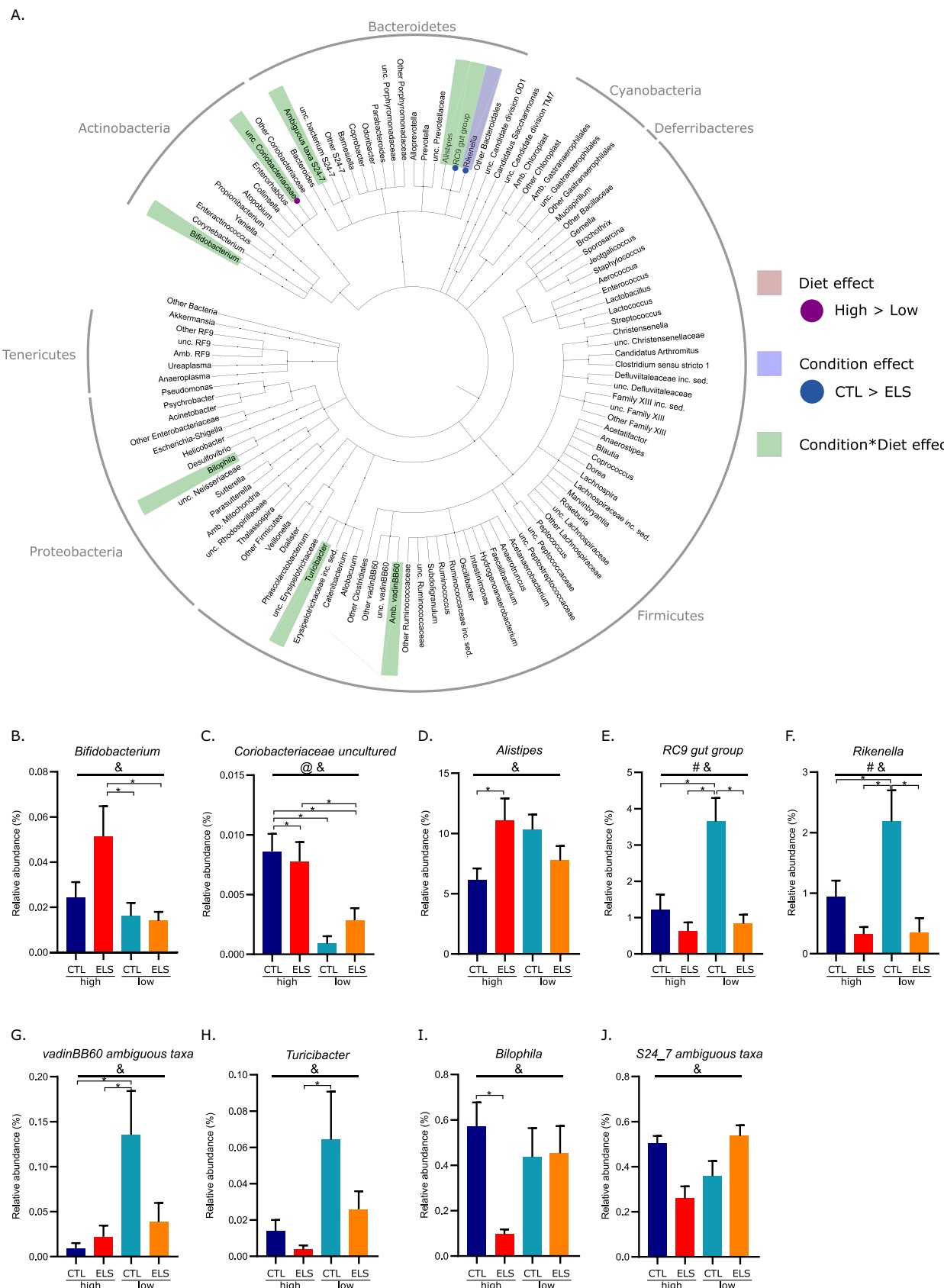

**FIG 3** Early-life stress and early dietary $\omega$-6/$\omega$-3 ratio affect the microbiota composition at P180 in interaction with each other. (A) Cladogram showing significant condition- and diet-mediated changes in the relative abundance of bacterial taxa at several taxonomic levels at P180. (B to J)

**TABLE 1** Significant condition and diet effects on bacterial taxa at several taxonomic levels at P42 and P180[a,b]

| Bacterial group | Taxonomic level | Age | Effect | F value | P value | q value | Litter correction (new F value) |
|---|---|---|---|---|---|---|---|
| Cyanobacteria | Phylum | P42 | Condition-diet | 5.134 | 0.005 | 0.050 | NA |
| Melainabacteria | Class | P42 | Condition-diet | 5.134 | 0.005 | 0.048 | NA |
| Gastranaerophilales | Order | P42 | Condition-diet | 5.134 | 0.005 | 0.068 | NA |
| Erysipelotrichia | Class | P42 | Diet | 14.974 | 0.000 | 0.009 | 11.532 |
| Erysipelotrichia | Class | P42 | Condition-diet | 6.277 | 0.002 | 0.033 | 5.057 |
| Erysipelotrichales | Order | P42 | Diet | 14.974 | 0.000 | 0.009 | 11.532 |
| Erysipelotrichales | Order | P42 | Condition-diet | 6.277 | 0.002 | 0.047 | 5.057 |
| Erysipelotrichaceae | Family | P42 | Diet | 14.974 | 0.000 | 0.009 | 11.532 |
| Erysipelotrichaceae | Family | P42 | Condition-diet | 6.277 | 0.002 | 0.044 | 5.057 |
| Allobaculum | Genus | P42 | Condition-diet | 5.733 | 0.003 | 0.033 | NA |
| Clostridiales FamilyXIII | Family | P42 | Condition-diet | 6.331 | 0.002 | 0.044 | NA |
| FamilyXIII incertae sedis | Genus | P42 | Condition-diet | 6.931 | 0.001 | 0.026 | NA |
| FamilyXIII_unc. | Genus | P42 | Condition-diet | 4.381 | 0.011 | 0.043 | NA |
| Clostridiales_other | Family | P42 | Condition-diet | 5.297 | 0.004 | 0.061 | NA |
| Clostridiales_other | Genus | P42 | Condition-diet | 5.297 | 0.004 | 0.033 | NA |
| Coprococcus | Genus | P42 | Condition | 2.985 | 0.045 | 0.148 | 4.302 |
| Odoribacter | Genus | P42 | Condition-diet | 4.327 | 0.011 | 0.043 | 4.327 |
| | | | | | | | |
| Bifidobacterium | Genus | P180 | Condition-diet | 4.027 | 0.014 | 0.036 | NA |
| Coriobacteriaceae_unc. | Genus | P180 | Diet | 24.924 | 0.000 | 0.001 | NA |
| Coriobacteriaceae_unc. | Genus | P180 | Condition-diet | 8.575 | 0.000 | 0.002 | NA |
| Alistipes | Genus | P180 | Condition-diet | 2.928 | 0.046 | 0.075 | NA |
| RC9 gut group | Genus | P180 | Condition | 10.670 | 0.002 | 0.074 | 4.950 |
| RC9 gut group | Genus | P180 | Condition-diet | 10.437 | 0.000 | 0.001 | 6.522 |
| Rikenella | Genus | P180 | Condition | 12.487 | 0.001 | 0.070 | 8.791 |
| S24-7_Amb.taxa | Genus | P180 | Condition-diet | 6.454 | 0.001 | 0.007 | 3.467 |
| VadinBB60_Amb.taxa | Genus | P180 | Condition-diet | 4.677 | 0.007 | 0.026 | NA |
| Turicibacter | Genus | P180 | Condition-diet | 3.601 | 0.022 | 0.043 | NA |
| Bilophila | Genus | P180 | Condition-diet | 4.429 | 0.009 | 0.027 | NA |

[a]GLMM $P < 0.05$ and $q < 0.1$.
[b]NA, not applicable; unc., uncultured.

mals fed the high-$\omega$-6/$\omega$-3 diet. For *S24-7* ambiguous taxa (*Bacteroidetes*) an interaction effect was found between ELS and diet at P180; however; *post hoc* testing did not reveal significant differences between the experimental groups.

To summarize, age had a large impact on microbiota species abundance; for example, the abundance of the phylum *Bacteroidetes* increased, while *Actinobacteria* and *Verrucomicrobi*a decreased with age. Next to the independent effects of ELS and dietary $\omega$-6/$\omega$-3 ratio on the abundance of several microbiota taxa, the majority of the changes in taxa abundance depended on the interaction of ELS and diet at both P42 and P180.

**Correlations between bacterial taxa and peripheral and central outcome parameters within the same mice.** We have recently reported that ELS exposure altered central and peripheral fatty acid profiles and impaired cognition in these animals (9). Exposure to the low-$\omega$-6/$\omega$-3 PUFA diet between P2 and P42 was able to protect against the ELS-induced cognitive deficits in adulthood but did not affect the metabolic alterations. In order to investigate if and how alterations in the microbiota might relate to these changes, we studied the correlation between several outcomes (behavior [Fig. 4C and D], metabolic parameters [Fig. 4A], levels of central [Fig. 4B] and peripheral [Fig. S2A and B] FA levels, and the relative abundance of bacterial groups at different taxonomic levels). Correlations with rho of $>0.7$ or rho of $<-0.7$ are reported in the text and in Tables S2 and S3.

**Bacterial taxa at P42 in relation to behavior in adulthood.** With regard to adult behavior, we detected a negative correlation between the P42 levels of two related

**FIG 3** Legend (Continued)
Bar graphs of detected interaction effects (condition-diet) for bacterial taxa at P180 (GLMM $P < 0.05$ and q $< 0.1$). #, main effect of condition; @, main effect of diet; &, interaction of condition-diet; ^, significant difference with Tukey *post hoc* test ($P < 0.05$); GLMM, general linear mixed model.

**TABLE 2** Significant age (P42 versus P180) effects on bacterial taxa at several taxonomic levels[a]

| Bacterial group | Taxonomic level | F value | P value | q value | Litter correction (New F value) |
|---|---|---|---|---|---|
| Actinobacteria | Phylum | 5.4757 | 0.0221 | 0.0487 | 3.8779 |
| Bacteroidetes | Phylum | 34.4424 | <0.0001 | <0,0001 | 34.5562 |
| Cyanobacteria | Phylum | 3.851 | 0.0537[b] | 0.0844 | NA |
| Proteobacteria | Phylum | 4.6601 | 0.0343 | 0.0629 | 2.0967 |
| Tenericutes | Phylum | 6.4105 | 0.0136 | 0.0374 | 2.4779 |
| Verrucomicrobia | Phylum | 23.9499 | 0.0000 | <0,0001 | 6.3759 |
| Coriobacteriia | Class | 4.7412 | 0.0328 | 0.0802 | 2.8819 |
| Bacteroidia | Class | 34.4424 | <0.0001 | <0,0001 | 34,5562 |
| Candidate division TM7_unc | Class | 4.4169 | 0.0392 | 0.0862 | 4.4169 |
| Melainabacteria | Class | 3.8439 | 0.0539[b] | 0.1078[b] | NA |
| Bacilli | Class | 17.3144 | <0.0001 | 0.0004 | NA |
| Firmicutes_other | Class | 35.0432 | <0.0001 | <0,0001 | NA |
| Betaproteobacteria | Class | 11.6267 | 0.0011 | 0.0040 | 7.9829 |
| Epsilonproteobacteria | Class | 5.2764 | 0.0246 | 0.0677 | NA |
| Mollicutes | Class | 6.4105 | 0.0136 | 0.0427 | 2.4779 |
| Verrucomicrobiae | Class | 23.9499 | <0.0001 | <0,0001 | 6.3759 |
| Micrococcales | Order | 6.3277 | 0.0142 | 0.0473 | 0.5026 |
| Coriobacteriales | Order | 4.7412 | 0.0328 | 0.0895 | 2.8819 |
| Bacteroidales | Order | 34.4424 | <0.0001 | <0.0001 | 34.5562 |
| Candidate division TM7_unc. | Order | 4.4169 | 0.0392 | 0.0980 | NA |
| Lactobacillales | Order | 18.7414 | <0.0001 | 0.0003 | NA |
| Firmicutes_other | Order | 35.0432 | 0.0001 | 0.0001 | NA |
| Burkholderiales | Order | 11.6324 | 0.0011 | 0.0054 | 7.9858 |
| Neisseriales | Order | 6.4864 | 0.0131 | 0.0473 | 6.2505 |
| Campylobacterales | Order | 5.2764 | 0.0246 | 0.0738 | 3.3948 |
| Anaeroplasmatales | Order | 7.4472 | 0.0080 | 0.0344 | 5.4174 |
| Verrucomicrobiales | Order | 23.9499 | 0.0001 | 0.0001 | 6.3759 |
| Corynebacteriaceae | Family | 4.1423 | 0.0456 | 0.0982 | NA |
| Micrococcaceae | Family | 6.3277 | 0.0142 | 0.0453 | 0.5026 |
| Coriobacteriaceae | Family | 4.7412 | 0.0328 | 0.0799 | 2.8819 |
| Porphyromonadaceae | Family | 7.8001 | 0.0067 | 0.0314 | NA |
| Rikenellaceae | Family | 5.4204 | 0.0228 | 0.0638 | 5.4204 |
| S24-7 | Family | 39.2163 | <0.0001 | <0.0001 | 38.3709 |
| Bacteroidales_other | Family | 6.2779 | 0.0146 | 0.0453 | NA |
| Candidate division TM7_unc. | Family | 4.4169 | 0.0392 | 0.0914 | NA |
| Gastranaerophilales_amb. taxa | Family | 4.8249 | 0.0314 | 0.0798 | 1.4402 |
| Staphylococcaceae | Family | 9.3853 | 0.0031 | 0.0167 | 4.5474 |
| Enterococcaceae | Family | 4.2443 | 0.0431 | 0.0965 | NA |
| Lactobacillaceae | Family | 18.6121 | 0.0001 | 0.0004 | NA |
| Christensenellaceae | Family | 30.1204 | <0.0001 | <0.0001 | NA |
| Family XIII | Family | 5.4217 | 0.0228 | 0.0638 | 5.0076 |
| Peptococcaceae | Family | 26.1085 | <0.0001 | <0.0001 | 20.4528 |
| Peptostreptococcaceae | Family | 7.5608 | 0.0076 | 0.0321 | 15.6048 |
| VadinBB60 | Family | 9.2728 | 0.0033 | 0.0167 | NA |
| Firmicutes_other | Family | 35.0432 | <0.0001 | <0.0001 | NA |
| Alcaligenaceae | Family | 11.6324 | 0.0011 | 0.0076 | 7.9858 |
| Neisseriaceae | Family | 6.4864 | 0.0131 | 0.0453 | 6.2505 |
| Helicobacteraceae | Family | 5.2764 | 0.0246 | 0.0656 | NA |
| Anaeroplasmataceae | Family | 7.4472 | 0.0080 | 0.0321 | 5.4174 |
| RF9_Amb. Taxa | Family | 7.1975 | 0.0091 | 0.0340 | 1.7373 |
| RF9_unc. | Family | 11.2451 | 0.0013 | 0.0080 | NA |
| Verrucomicrobiaceae | Family | 23.9499 | <0.0001 | 0.0001 | 6.3759 |
| Enteractinococcus | Genus | 5.6163 | 0.0206 | 0.0555 | 0.0768 |
| Collinsella | Genus | 5.3448 | 0.0237 | 0.0625 | 2.8641 |
| Enterorhabdus | Genus | 17.137 | 0.0001 | 0.0007 | 21.7296 |
| Coriobacteriaceae_other | Genus | 7.912 | 0.0064 | 0.0246 | 1.11 |
| Parabacteroides | Genus | 14.1533 | 0.0003 | 0.0024 | 7.0403 |
| RC9 gut group | Genus | 12.5971 | 0.0007 | 0.0040 | 11.1118 |
| Rikenella | Genus | 4.7196 | 0.0332 | 0.0797 | 5.1315 |
| S24-7 ambiguous_taxa | Genus | 6.3066 | 0.0143 | 0.0425 | 8.1212 |
| S24-7_unc. | Genus | 39.2924 | <0.0001 | <0.0001 | 39.2924 |

**TABLE 2** (Continued)

| Bacterial group | Taxonomic level | F value | P value | q value | Litter correction (New F value) |
|---|---|---|---|---|---|
| S24-7_other | Genus | 23.8128 | <0.0001 | 0.0001 | NA |
| Bacteroidales_other | Genus | 6.2779 | 0.0146 | 0.0425 | NA |
| Candidate division TM7_unc. | Genus | 4.4169 | 0.0392 | 0.0920 | NA |
| Gastranaerophilales_amb. taxa | Genus | 4.8249 | 0.0314 | 0.0770 | 1.4402 |
| Jeotgalicoccus | Genus | 4.9322 | 0.0296 | 0.0743 | 2.3883 |
| Staphylococcus | Genus | 8.5913 | 0.0046 | 0.0205 | 4.7786 |
| Enterococcus | Genus | 4.2443 | 0.0431 | 0.0990 | NA |
| Lactobacillus | Genus | 18.6121 | 0.0001 | 0.0005 | NA |
| Lactococcus | Genus | 8.139 | 0.0057 | 0.0246 | NA |
| Christensenellaceae_unc. | Genus | 31.6138 | <0.0001 | <0.0001 | 30.8904 |
| Incertae sedis | Genus | 13.8341 | 0.0004 | 0.0025 | 11.1466 |
| Family XIII_unc. | Genus | 10.2869 | 0.0020 | 0.0099 | 9.7733 |
| Blautia | Genus | 10.0027 | 0.0023 | 0.0109 | 0.014 |
| Coprococcus | Genus | 7.111 | 0.0095 | 0.0310 | 17.8824 |
| Roseburia | Genus | 14.1223 | 0.0004 | 0.0024 | 27.0365 |
| Lachnospiraceae_unc. | Genus | 7.9302 | 0.0063 | 0.0246 | NA |
| Peptococcus | Genus | 7.9786 | 0.0062 | 0.0246 | NA |
| Peptococcaceae_unc. | Genus | 24.4036 | <0.0001 | 0.0001 | 19.6722 |
| Peptostreptococcaceae_unc. | Genus | 7.5608 | 0.0076 | 0.0282 | 15.6048 |
| Anaerotruncus | Genus | 28.9301 | <0.0001 | <0.0001 | 44.3953 |
| Intestinimonas | Genus | 17.3829 | 0.0001 | 0.0007 | 12.9786 |
| vadinBB60_amb. taxa | Genus | 5.7889 | 0.0188 | 0.0520 | NA |
| vadinBB60_unc. | Genus | 34.9198 | <0.0001 | <0.0001 | NA |
| vadinBB60_other | Genus | 28.1352 | <0.0001 | <0.0001 | NA |
| Incertae sedis | Genus | 13.6205 | 0.0004 | 0.0026 | NA |
| Turicibacter | Genus | 7.0612 | 0.0098 | 0.0310 | 2.4219 |
| Erysipelotrichaceae_unc. | Genus | 6.069 | 0.0162 | 0.0461 | NA |
| Firmicutes_other | Genus | 35.0432 | <0.0001 | <0.0001 | NA |
| Rhodospirillaceae_unc. | Genus | 7.2704 | 0.0088 | 0.0306 | 4.1117 |
| Parasutterella | Genus | 11.6324 | 0.0011 | 0.0058 | 7.9858 |
| Neisseriaceae_unc. | Genus | 6.4864 | 0.0131 | 0.0403 | 6.2505 |
| Bilophila | Genus | 26.445 | <0.0001 | <0.0001 | NA |
| Helicobacter | Genus | 5.2764 | 0.0246 | 0.0633 | NA |
| Anaeroplasma | Genus | 7.4472 | 0.0080 | 0.0289 | 5.4174 |
| RF9_amb. taxa | Genus | 7.1975 | 0.0091 | 0.0307 | 1.7373 |
| RF9_unc. | Genus | 11.2451 | 0.0013 | 0.0066 | NA |
| Akkermansia | Genus | 23.9499 | <0.0001 | 0.0001 | 6.3759 |

[a]GLMM $P < 0.05$ and $q < 0.1$.
[b]Trend.
[c]NA, not applicable; unc., uncultured.

Bacteroidetes taxa, Porphyromonadaceae and Odoribacter, and performance on the object location task (OLT) (rho = −0.7 and rho = −0.73, respectively) (Fig. 4C and D). No correlations were detected for the other parameters related to behavior.

**Bacterial taxa at P42 in relation to metabolic outcome parameters at P42.** The abundance of several bacterial species at P42 correlated with specific P42 metabolic outcomes (Fig. 4A; Table S2). Namely, the phylum Bacteroidetes and order Bacteroidales were negatively correlated with the amount of inguinal fat (rho = −0.77 for both). Taxa of the Bacteroidetes phylum, Porphyromonadaceae and Odoribacter, were positively correlated with body weight (rho = 0.73 for both) (Fig. 4E). Several taxa within the Proteobacteria phylum—Enterobacteriales, Enterobacteriaceae, and the Escherichia-Shigella group(rho = 0.71) —as well as taxa within the Firmicutes phylum—Clostridiaceae 1 and Clostridium sensu stricto 1 (rho = 0.85 for both) and Marvinbryantia (rho = 0.78)—were positively correlated with plasma leptin levels. The genus Christensenella and Bacteroidetes S24-7 ambiguous taxa were negatively correlated with leptin levels (rho = −0.71 and −0.83, respectively). The Bacteroidetes S24-7 and S24-7 Unc. showed a negative correlation with the amount of white fat in mice (rho = −0.72 for both). There were no correlations between bacterial species at P180 and metabolic outcomes at P180.

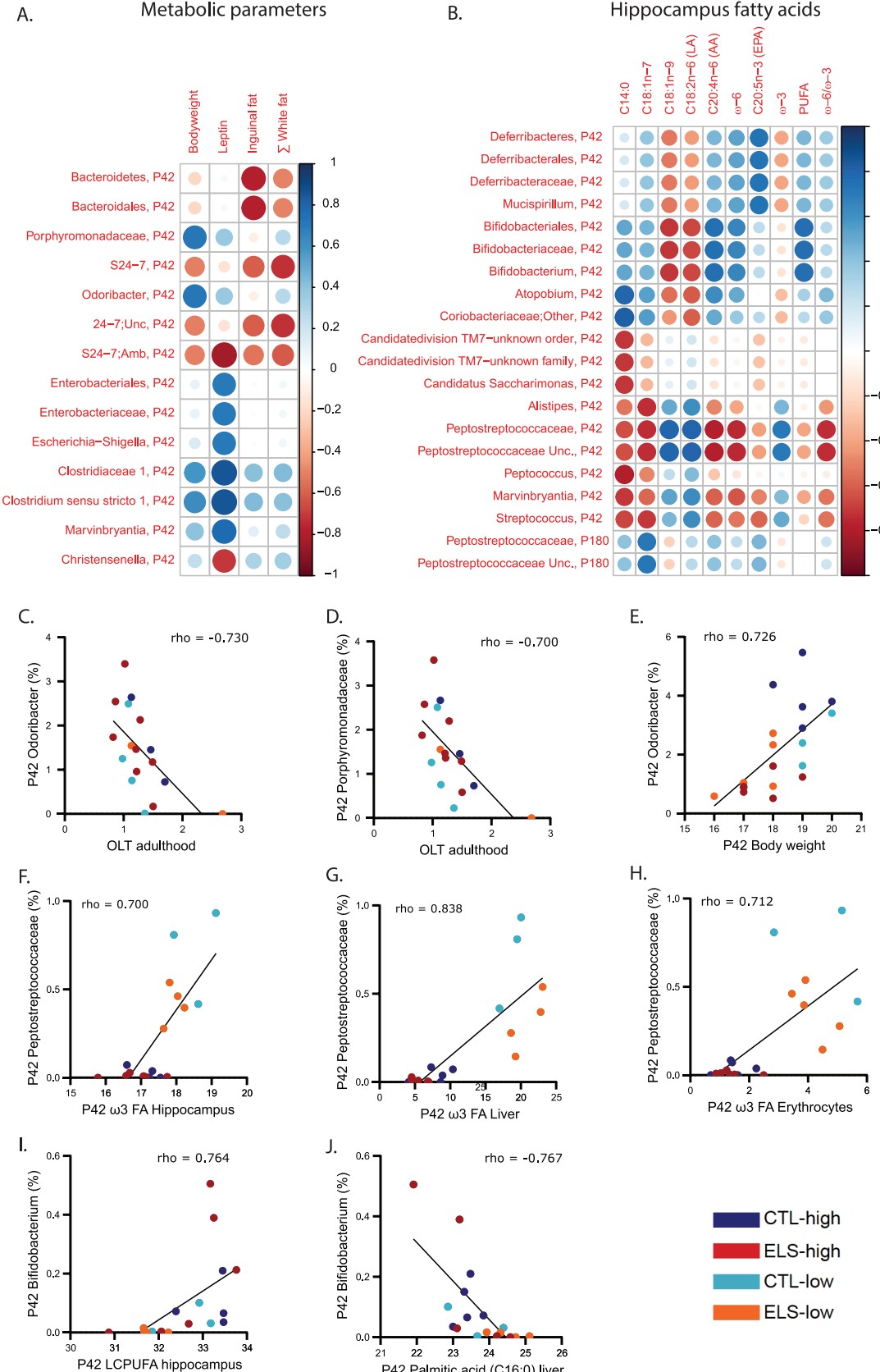

**FIG 4** Bacterial taxa are correlated with several peripheral and central outcome parameters within the same mice. (A) Correlation heatmap between bacterial taxa at P42 and metabolic outcome parameters at P42; (B) correlation heatmap between bacterial taxa

**Bacterial taxa at P42 in relation to fatty acid levels in the hippocampus, erythrocytes, and liver at P42.** We detected multiple strong correlations between bacterial taxa and fatty acid levels in the hippocampus, erythrocytes, and liver (Fig. 4B; Table S3). From the *Firmicutes* phylum the *Peptostreptococcaceae* family negatively was correlated with the $\omega$-6/$\omega$-3 ratio in the hippocampus, erythrocytes, and liver (rho values of $-0.76$, $-0.71$, and $-0.77$, respectively). In agreement with this, *Peptostreptococcaceae* were positively correlated with $\omega$-3 fatty acid levels in all three tissues (rho > 0.7 for all three tissues) (Fig. 4F to H). This effect was mostly due to a lack of abundance of *Peptostreptococcaceae* in mice fed the high-$\omega$-6/$\omega$-3 diet versus those fed the low-$\omega$-6/$\omega$-3 diet. Next, the *Lactobacillaceae* family, also from the *Firmicutes* phylum, were positively correlated with the monounsaturated fatty acid/linoleic acid (LA/ALA) ratio in erythrocytes and liver (rho = 0.73 for both tissues). Within the *Actinobacteria* phylum the *Bifidobacterium* lineage (from order, family until genus level) was positively correlated with the amount of LCPUFA in the hippocampus (rho = 0.82) (Fig. 4I). In line with the GLMM results, ELS mice fed the high-$\omega$-6/$\omega$-3 diet contained the highest relative abundance of *Bifidobacterium* (Fig. 4I), while an inverse correlation was detected between *Bifidobacterium* and the amount of palmitic acid in the liver (rho = 0.767) (Fig. 4J). We detected very few correlations between P180 bacterial species and fatty acid levels at P180 (Fig. 4B; Fig. S2A and B; Tables S2 and S3).

In summary, when relating the microbial taxon abundance to the earlier-reported central and peripheral outcome measures (9), several correlations became apparent. For example, we detected (i) a negative correlation between the abundance of two *Bacteroidetes* taxa and adult behavior (*Porphyromonadaceae* and *Odoribacter*) and several correlations between bacterial taxa (ii) and metabolic outcomes (i.e., white fat mass and circulating leptin) and (iii) with peripheral and central fatty acid levels, mostly at P42.

## DISCUSSION

We have previously shown that an early dietary intervention with a reduced $\omega$-6/$\omega$-3 PUFA (LA/ALA) ratio protects against the ELS-induced cognitive deficits without affecting the metabolic alterations (9). While the relation between stress, nutrition, and the gut microbiota has been gaining increased attention over the recent years (13, 32), the specific mechanisms of such dietary interventions are not well understood. Here, we demonstrate that, while $\alpha$-diversity is impacted by age only, chronic ELS during the first week of life (P2 to P9) and early dietary $\omega$-6/$\omega$-3 ratio, mostly in interaction with each other, modulate $\beta$-diversity and the relative abundance of bacterial groups at several taxonomic levels in the short and long term.

We will first discuss the microbiota diversity and composition across age, then elaborate on the short- and long-term impact of ELS and early dietary $\omega$-6/$\omega$-3 ratio on different microbiota parameters, and lastly, relate the microbial taxon abundance to earlier-reported central and peripheral outcome measures from the same cohort of mice (9).

**The microbiota across age.** The phylogenetic diversity within samples ($\alpha$-diversity) increased with age from weaning (P21) up to adulthood and, as expected, with only a relatively small difference between P42 and P180 samples in terms of the number of detected species. This is in line with the previously described total amount of species across these ages (43, 44), while a decrease in $\alpha$-diversity has been described in late adulthood or old age, which was associated with increased presence of diseases and medication (45). The sample size at weaning age (P21) was relatively low, and even though the methodology that was used was reliable and sensitive enough to pick up

**FIG 4** Legend (Continued)

at P42 and P180 and fatty acid levels in the hippocampus at P42 and P180, respectively ($-1 <$ Spearman's rho $< 1$). (C to J) Correlation plots between relative abundance of selected bacterial taxa and behavioral, metabolic, and/or fatty acid outcomes. (C) P42 *Odoribacter* and adult OLT performance. (D) P42 *Porphyromonadaceae* and OLT performance. (E) *Odoribacter* and body weight. (F) P42 *Peptostreptococcaceae* and P42 $\omega$-3 FA hippocampus. (G) P42 *Peptostreptococcaceae* and P42 liver $\omega$-3 FA. (H) P42 *Peptostreptococcaceae* and P42 erythrocyte $\omega$-3 FA. (I) *Bifidobacterium* and hippocampal LCPUFA levels. (J) *Bifidobacterium* and liver palmitic acid ($C_{16:0}$).

age-related changes in $\alpha$-diversity (sequencing depth of over 20,000 sequences for all three ages), we will further focus the discussion on our findings comparing adolescent (P42) and adult (P180) microbiota composition. The composition of the gut microbiota in terms of its relative species abundance is affected by age. At both P42 and P180, *Bacteroidetes* and *Firmicutes* are the two most abundant phyla in all experimental groups, which is in line with other rodent and human microbiota profiles (46). When comparing these two ages, we observed multiple changes in the composition of the fecal microbiota, mainly consisting of a reduction in the phyla *Actinobacteria* and *Verrucomicrobia* (which includes the genus *Akkermansia*) and an increase of *Bacteroidetes* in adulthood. In particular, in P180 samples compared to P42 samples, we observed lower abundance of members of the phylum *Actinobacteria* (*Coriobacteriaceae* and *Enterorhabdus*), of which *Bifidobacterium* is a genus, and multiple members of the *Firmicutes* order *Clostridiales*. Also, we observed higher abundances of members of the phylum *Proteobacteria*, such as the genus *Parasutterella*, in P180 samples compared to P42 samples. In line with our comparative analyses between ages in mice, there is evidence for age-dependent changes in the microbiome from human literature. While most studies to date aimed at comparing gut microbiota of children between 0 and 2 years old with those of adults or the elderly (47), very few have included adolescent groups. However, based on Agans et al., in line with our findings, adolescents can easily be separated from adults based on the relative species abundance and because, in particular, adolescent microbiota consist of a relatively lower abundance of the genus *Sutterella* and relatively higher abundance of *Bifidobacterium* and *Clostridium* (48, 49). Further work is needed, in both rodent and human cohorts, to be able to understand the age-related changes in microbiota composition in more detail and if and how each age group might be differently sensitive to stress exposure, diet, or other environmental challenges.

**Short- and long-term impact of early-life stress and early diet on microbiota composition.** We will here first discuss the effects of ELS on microbiota $\alpha$- and $\beta$-diversity and species abundance and then the specific effects of the different dietary $\omega$-6/$\omega$-3 ratios on these parameters, as well as the interaction of $\omega$-6/$\omega$-3 ratio with ELS exposure.

**Short- and long-term impact of early-life stress on microbiota composition.** In this study, ELS exposure did not affect $\alpha$-diversity, and ELS-induced effects on $\beta$-diversity at P42 and P180 depended on the early dietary $\omega$-6/$\omega$-3 ratio. In line with our findings, a multihit ELS model did not alter $\alpha$-diversity in adult mice (50), while others have reported an ELS reduction in $\alpha$-diversity in rats, via the limited bedding and nesting (LBN) model at weaning (51) or maternal separation (MS) in adulthood (52). Similarly, differences in phylogenetic $\beta$-diversity have been reported in some (51) but not all ELS studies (42, 50). Thus, type of ELS model, outcome age, and species seem to greatly impact the effects of early life adversity on the microbial $\alpha$- and $\beta$-diversity. In general, a less diverse microbiome is thought to be less resilient to external perturbations due to the loss of functional redundancy of the present species and therefore possibly be less healthy (53). However, whether health outcomes are positive or negative likely depends on the actual composition of the community.

Few bacterial species were affected by ELS regardless of the early dietary $\omega$-6/$\omega$-3 ratio. At P42, ELS reduced the abundance of the genus *Coprococcus*, part of the *Lachnospiraceae* family. This is in line with the reduction in *Coprococcus* found at weaning in ELS-exposed rats, via the LBN paradigm (51). *Lachnospiraceae* and *Coprococcus* have been defined as major butyrate-producing bacterial groups in both rodents and humans (54, 55), which suggests that ELS could affect butyrate levels via affecting these taxa. In adulthood (P180) the genera RC9 gut group and *Rikenella*, both part of the *Rikenellaceae* family and *Bacteroidetes* phylum, were lastingly reduced by ELS. Similarly, MS in rats has been shown to reduce abundance of *Rikenella*, which is also correlated with stress-induced corticosterone plasma levels in MS-exposed rats (42). *Rikenella* is a well-known sugar fermenter, and it has been suggested that stress can reduce the availability of sugars in the gut (56), possibly leading to a decrease of bacteria involved in processing of sugars.

In summary, while not impacting $\alpha$-diversity, ELS affected $\beta$-diversity dependent on early diet. The implications of the above-mentioned ELS-induced alterations within the

relative abundance of bacterial groups are not yet well understood but nevertheless can impact the functionality of the gut microbiota, possibly via affecting butyrate and sugar metabolism.

**Effect of the early dietary ω-6/ω-3 ratio and its interaction with early-life stress in the short and long term.** We have previously reported, within this same cohort, a rescue effect of the low-ω-6/ω-3 diet on the ELS-induced cognitive impairments as well as alterations in hippocampal brain plasticity, namely, a reversal of the ELS reduction in adult neurogenesis and the ELS-increase in the phagocytic marker of microglia, without affecting the ELS-mediated metabolic changes (9). This allows us to not solely discuss effects of ELS and early diet on the microbiota, but also relate the observed changes to earlier-described ELS-induced alterations, which were performed within the same mice cohort.

While there was no main effect of ELS or early dietary ω-6/ω-3 ratio on phylogenetic β-diversity independent of one another, these were apparent when the interactions between the two early-life conditions were taken into consideration. In addition, when looking at the relative abundance of microbial species at taxonomic levels, while some main effects of early dietary ω-6/ω-3 ratio were detected, here as well, most diet effects were dependent on previous ELS exposure. Some immediate and long-lasting effects of the diet were observed, though. For example, mice fed the low-ω-6/ω-3 ratio diet from P2 to P42, exhibited a reduction in *Erysipelotrichia* lineage down to the *Erysipelotrichaceae* family (*Firmicutes*) compared to mice fed the high-ω-6/ω-3 ratio diet. These taxa have been reported to be increased in obese individuals notoriously consuming diets with excess of ω-6 fatty acids (57), pointing toward the idea that dietary ω-6/ω-3 ratio is an important modulator of these specific bacteria and their balance. Similarly, ω-3 LCPUFA supplementation has been shown to lead to a decrease of the *Firmicutes* phylum (33, 58, 59) and restoration of the *Firmicutes/Bacteroidetes* ratio, often reported to be higher under pathological conditions such as obesity and inflammatory bowel syndrome (IBS) (60, 61).

Next to the independent effects of ELS and dietary ω-6/ω-3 ratio, their interaction is particularly interesting to gain further insight in how the diet might exert its protective effect on the ELS-induced deficits. For example, directly after the end of the dietary intervention at P42, specifically control mice fed the low-ω-6/ω-3 diet exhibited an increased abundance of several *Clostridia* members compared to those fed the diet with the high ω-6/ω-3 ratio. These taxa belong to the phylum *Firmicutes* and order *Clostridiales*, which are known for their involvement in the production of butyrate (62, 63). Such modulation is in line with the fact that ω-3 fatty acid supplementation can indeed lead to increased bacterially derived butyrate (34, 54). Short-chain fatty acids (SCFA) such as butyrate, propionate, and acetate are bacterially derived metabolites of fibers and carbohydrate that have been suggested to be key for mental health (15), for example, by increasing central brain-derived neurotrophic factor (BDNF) production (64, 65) and modulation of microglial maturation and functionality (66). As mentioned above, the low-ω-6/ω-3 diet was able to prevent ELS-induced alterations in hippocampal plasticity, including microglial morphology and phagocytic capacity (9), raising the question if this could possibly be related to an increase in bacterially derived butyrate by the diet in ELS-exposed animals specifically. Next, ELS exposure reduced the abundance of the *Bacteroidetes* genus *Odoribacter* in animals fed the high-ω-6/ω-3 diet but not in animals fed the low-ω-6/ω-3 diet. *Odoribacter* is a known producer of acetate, propionate, and butyrate (67), and decreased *Odoribacter* may affect host inflammation via reduced SCFA availability. It will be interesting to see in follow-up studies whether, indeed, our conditions lead to altered levels of SCFAs and, in particular, butyrate. At P180, *Bifidobacterium* and *Bilophila* also exhibited diet-ELS interaction effects. Levels of *Bifidobacterium* were specifically increased, while levels of *Bilophila* were decreased in ELS-exposed mice fed the high-ω-6/ω-3 diet compared to the other experimental groups. As further discussed under "Abundance of Microbiota Species in Relation to Central and Peripheral Outcomes," below, *Bifidobacterium* was correlated positively

with hippocampal LCPUFAs and negatively with the saturated fatty acid palmitic acid. *Bifidobacteria* are Gram-positive, anaerobic bacteria that belong to the phylum *Actinobacteria* and are among the first bacteria to colonize the gastrointestinal tract, with a high abundance during the first stages of life that decreases over time (43). *Bilophila* is a bile-tolerant bacterium, and high levels have been associated with inflammation and dietary lipids (68, 69), as well as with individuals suffering from severe malnutrition (70). Whether the observed reduction in the context of ELS and early dietary PUFAs contributes to the adiposity and neuroinflammatory changes induced by these conditions remain to be determined.

Notably, several of the detected diet-mediated changes are in line with literature showing beneficial effects of dietary $\omega$-3 supplementation on brain and metabolism (40, 71) and suggest that the diet-induced protective effects might be partly modulated by the observed changes in microbiota. We hypothesize that lowering $\omega$-6/$\omega$-3 early in life contributes to a stable and diverse microbiota, thereby affecting sensitive developmental processes that could impact the later-life health status.

Important to note is that within the current study, all animals were on a lifelong synthetic diet, enabling us to control for the source and proportion of its ingredients. Such synthetic diets, also referred to as "refined," contain mostly insoluble fibers such as cellulose (Table S4), which distinguishes them from the regular chow diets containing both soluble and insoluble fibers. These dietary conditions likely impact microbiota composition since distinct bacterial species are involved in the fermentation of soluble versus insoluble fibers (72, 73). While this does not affect the differences observed between groups in the current study, as all experimental groups were exposed to the synthetic diet, it is important to bear this in mind when comparing the current findings to existing literature (74).

**Abundance of microbiota species in relation to central and peripheral outcomes.** We studied how the bacterial changes correlate with the previously published ELS- and diet-mediated differences in cognitive abilities, metabolic alterations, and central and peripheral fatty acid profiles analyzed in this same cohort (9). As mentioned in "Short- and Long-Term Impact of Early-Life Stress on Microbiota Composition," above, we found a negative correlation between adult performance on a spatial memory task and the levels of the *Bacteroidetes* family *Porphyromonadaceae* and its genus *Odoribacter* at P42 and not at P180. Similarly, an increased abundance in both taxa has been described in aged mice (75), and specifically, *Porphyromonadaceae* has been shown to be negatively correlated with cognitive dysfunction in humans (76, 77), suggesting that a dysregulation of these taxa might be key in modulating cognitive functions. We have previously reported that ELS exposure leads to a lifelong reduction in white fat mass and circulating leptin (78), while these ELS-induced effects were not modulated by the diet (9). When studying the correlation of the bacterial profile with the metabolic outcomes (body weight, fat mass, and leptin), we found a positive correlation of *Porphyromonadaceae* and *Odoribacter* with bodyweight at P42, which is in line with a previous report showing that their abundance is increased in high-fat diet (HFD)-exposed mice (79). In addition, the phylum *Bacteroidetes* and multiple of its taxa (e.g., S24-7) were negatively correlated with the amount of white fat mass. Notably, an unidentified taxon from the S24-7 family has been reported to be affected by early-life supplementation of synbiotics that protected against diet-induced obesity in adult mice (80). Indeed, high levels of *Bacteroidetes* and some of its taxa are associated with a healthy non-Western diet, while lower levels are associated with a Western-style diet (32, 81). Finally, there was a positive correlation between several taxa within the *Proteobacteria* phylum and plasma leptin levels. Importantly, changes in the *Proteobacteria* have been associated with HFD in mice and humans, where leptin levels are dysregulated as well (57, 82, 83). Also, *Bacteroidetes* S24-7-ambiguous taxa and the *Firmicutes* genus *Christensenella* were negatively correlated with plasma leptin, and both were previously associated with reduction in body weight or adiposity in mice (80, 84), suggesting that these bacteria might be particularly sensitive to conditions

with altered leptin and fat mass. Lastly, we detected multiple strong correlations between bacterial species and specific fatty acid levels in the hippocampus, liver, and erythrocytes. To name a few examples, there was a negative correlation between the P42 hippocampal, liver, and erythrocyte $\omega$-6/$\omega$-3 ratio and the relative abundance of *Peptostreptococcaceae* (*Firmicutes*) at P42. In agreement, *Peptostreptococcaceae* positively was correlated with $\omega$-3 FA levels in all three tissues. Interestingly a lifelong $\omega$-3 PUFA supplementation starting prenatally led to decreased levels of *Peptostreptococcaceae* compared to chow-fed or $\omega$-3-deficient mice (85). Such discrepancy is mostly likely due to the length and type of the dietary intervention. Next, the *Bifidobacterium* genus, which has been established to be increased by diets high in $\omega$-3 fatty acids (34, 85), was positively correlated with the amount of hippocampal PUFAs, while an inverse correlation was detected between *Bifidobacterium* and the most common saturated FA, palmitic acid ($C_{16:0}$), in the liver. Several positive functions have been attributed to bifidobacteria, such as degradation of nondigestible carbohydrates, production of vitamin B, antioxidants, stimulation of the immune system, and increasing butyrate levels via cross-feeding (86, 87). While the above-mentioned relations are, of course, of descriptive nature, they give us a lead for future investigations to better understand which processes might be most impacted by microbiota changes and via which routes the microbiota changes could be involved in the observed ELS- and diet-induced effects.

**Limitations of our study.** While our study presents some unique strengths, such as the experimental design and the unique combination of ELS and early diet on short- and long-term effects, it also presents some limitations. First, our study lacks the inclusion of female mice. As mentioned earlier, this study is the follow-up of a large investigation which encompassed both cognitive and metabolic readouts. This original study was designed to include males, as we had shown previously that the ELS model used affects cognitive function and neurogenesis primarily in males (7), and also, the reported effects of early diet with PUFAs on the long-term effects of ELS (9). The emerging evidence for the sex differences in the response to ELS (88–91), dietary interventions (92), and early-life nutrition (93) as well as in gut microbiota across the life span (94, 95) warrant studying the differential effects of ELS and dietary manipulations on the microbiome in both males and females in future experiments.

It is important to note that in our study we used the QIIME v.1.9.0 pipeline (96), with which processing of sequencing data is performed based on operational taxonomic units (OTUs). We are aware of the new developments in the field of rRNA 16S amplicon sequencing, which has moved toward using sequence variants (aSVs) for data analysis (97) because OTUs tend to inflate, in particular, measures of $\alpha$-diversity (98). In order to mitigate the potential inflation, we have now filtered out low-abundance OTUs (<0.002%) (99), and there is evidence that when assessing concatenated data at the genus level, which is what we did in the current study, there is no large difference between genus level data originating from OTUs or from aSV data (97). In fact, in this study, the analysis of phylogenetic $\beta$-diversity and the significance of taxa have been studied at the genus level or above. Taking this all together, we are confident about the reliability of our analyses.

In conclusion, we show that exposure to ELS during the first postnatal week and the $\omega$-6/$\omega$-3 ratio of the early diet from P2 to P42, especially in interaction with one another, affect the gut microbiota of male mice in the long term. These data give novel insights into the complex interaction between ELS, early dietary $\omega$-3 availability, and the gut microbiota across ages and provide a basis for future studies addressing the causal relationship between the alterations in microbiota, the ELS-induced deficits, and diet, as well as for noninvasive (nutritional) interventions targeting the microbiota to protect against and/or reverse the ELS-induced deficits.

## MATERIALS AND METHODS

**Animals.** In the current study, we describe microbiome data from the same mice as from our previous publication (9). In brief, male (6 weeks old) and primiparous female (8 weeks old) C57BL/6J mice were purchased from Harlan Laboratories B.V. (Venray, the Netherlands). After arrival at the animal

facility, the mice were put on a synthetic AIN-93G diet (Ssniff-Spezialdiäten GmbH, Germany) (100) and housed in a controlled environment (temperature, 22 $\pm$ 1°C; humidity, 55% $\pm$ 5%) with *ad libitum* food and water, under a 12-h:12-h light-dark cycle schedule (lights on at 8 a.m.). After 2 weeks of acclimatization, mice were bred in-house by housing two females with one male for 1 week in a type II long cage. Subsequently, females were housed in single-sex pairs for another week, and after that, pregnant females were housed individually in a standard cage (type I short cage) covered with a filter top. Females were monitored daily, between 9 and 10 a.m., for the birth of pups. When a litter was detected, the previous day was designated the day of birth (postnatal day 0 [P0]). At P2, dams with litter were randomly assigned to control (CTL) or ELS conditions (see "Chronic Early-Life Stress Exposure," below) and to one of the experimental diets (high- or low-$\omega$-6/$\omega$-3 ratio diet; see "Experimental Diets," below). At P21, offspring were weaned, and male offspring were housed in groups (littermates; 2 or 3 animals per cage) in type II long cages with a standard amount of bedding material. Mice were kept on their respective diet until P42, after which all groups were switched to standard semisynthetic diet (AIN93M) (100) until end of the study. 16S rRNA was sequenced and analyzed from fecal samples at P21, P42, and P180. The four experimental groups are control mice fed a diet with a high $\omega$-6/$\omega$-3 ratio, (15)-CTL-high; ELS-exposed mice fed a high $\omega$-6/$\omega$-3 ratio, (15)-ELS-high; control mice fed a diet with a low $\omega$-6/$\omega$-3 ratio, (1.1)-CTL-low; and ELS mice fed a diet with a low $\omega$-6/$\omega$-3 ratio, (1.1)-ELS-low. The sample size per group and per age was as follows: P21: CTL-high $n = 3$, ELS-high $n = 5$, CTL-low $n = 5$, ELS-low $n = 5$; P42: CTL-high $n = 9$, ELS-high $n = 14$, CTL-low $n = 7$, ELS-low $n = 7$; P180: CTL-high $n = 11$, ELS-high $n = 11$, CTL-low $n = 10$, ELS-low $n = 9$.

All experimental procedures were approved by the Animal Welfare Body of the University of Amsterdam and the Central Authority for Scientific Procedures on Animals (Centrale Commissie Dierproeven [CCD]) in compliance with Dutch legislation and the principles of good laboratory animal care following the European Union directive for the protection of animals used for scientific purposes.

**Chronic early-life stress exposure.** We used the chronic ELS model, based on the limited bedding and nesting (LBN) stress paradigm as described before by our group and others (5, 7, 9). The LBN paradigm induces fragmentation of maternal care, which results in chronic stress in the pups. At P2, litters were culled to six pups per litter (sex ratio, m:f of 3:3 or 4:2) without cross-fostering, randomly assigned to CTL or ELS conditions. In ELS cages, the bottom was covered with a small amount of sawdust bedding, and a fine-gauge stainless steel mesh was placed 1 cm above the cage floor. Half a square piece of cotton nesting material (2.5 by 5 cm; Technilab-BMI, Someren, the Netherlands) was placed on top of the mesh. Control cages were equipped with standard amounts of sawdust bedding and nesting material (one square piece of cotton nesting material (5 by 5 cm). Cages were equipped with food and water *ad libitum* and covered with a filter top. Throughout all procedures, manipulation was kept to a minimum to avoid handling effects, and animals were left undisturbed until P9. On the morning of P9, body weight of the dams and pups and the consumed amount of food and/or water were measured; these data can be found in our previous publication (9). From P9 onward, all animals were housed in cages equipped with a standard amount of nesting and bedding material.

**Experimental diets.** Experimental diets were provided from P2 onward to dams with litter, and after weaning (P21), offspring were kept on their respective diet until P42. During lactation, fatty acid composition of the maternal diet, in particular, LA and DHA, is reflected in milk fatty acid composition (100). The two experimental diets (Ssniff-Spezialdiäten GmbH, Soest, Germany) were semisynthetic, differing only in LA/ALA ratio, which was either high (15) or low (1.1). The diets were isocaloric and contained a macro- and micronutrient composition according to the AIN93-G purified diets for laboratory rodents (100) (Table S1).

**Fecal sample collection, DNA extraction, and sequencing.** Fresh fecal samples were collected during a brief handling moment of approximately 2 min from three separate age cohorts, P21, P42, and P180. One or two pellets per animal were snap-frozen and stored at $-80$°C until further analysis.

DNA extraction from these samples was performed with a QIAamp DNA stool minikit (Qiagen) according to the manufacturer's protocol except for the addition of two bead-beating steps. To 0.2 to 0.3 g of fecal sample, 300 mg of 0.1-mm glass beads together with 1.4 mL of stool lysis buffer (ASL) (lysis) buffer were added. On this suspension the first bead-beating step was applied for 3 $\times$ 30 s (FastPrep-24 instrument program v.5.5). After addition of the InhibitEX tablet, the second bead-beating step was applied for 3 $\times$ 30 s (FastPrep-24 instrument program v.5.5) to homogenize the sample. Following each bead-beating step, samples were cooled for 5 min on ice. Extracted DNA purity was checked using the NanoDrop spectrophotometer (Thermo Fisher Scientific, Inc.), whereas DNA quality and concentration were measured using the Quant-iT 193 double-stranded DNA (dsDNA) BR assay kit (Invitrogen). DNA aliquots were stored at $-20$°C until use.

On the purified fecal DNA extracts primers, Bact-0341F (5'-CCTACGGGNGGCWGCAG-3') and Bact-0785R (5'-GACTACHVGGGTATCTAATCC-3') (101) were used to amplify the V3-V4 regions of the bacterial 16S rRNA gene, and the generated amplicons were subsequently sequenced on an Illumina MiSeq instrument as described previously (102).

**Sequencing analysis.** Sequencing data were analyzed using the Quantitative Insights Into Microbial Ecology (QIIME) v.1.9.0 pipeline (96). Sequences with mismatched primers were discarded. Quality control filters were set to retain sequences with a length between 200 and 1,000 bases, a mean sequence quality score of >15 in a five-nucleotide window, and no ambiguous bases. The filtered sequences were grouped into operational taxonomic units (OTUs) by *de novo* OTU picking using the USEARCH algorithm (103) at 97% sequence identity. Subsequently, the Ribosomal Database Project classifier (RDP) (104) was applied to assign taxonomy to the representative sequence (i.e., the most abundant sequence) of each OTU by alignment to the SILVA rRNA database (release v.1.1.9) (105). ChimeraSlayer (106) was applied,

as part of QIIME, to filter for chimeric sequences, and these were excluded from all downstream analyses. Representative OTUs were aligned using PyNAST (96) and used to build a phylogenetic tree with FastTree (107). OTUs that could not be aligned with PyNAST, singletons, and low-abundance OTUs with a relative abundance of <0.002% were excluded to reduce inflation by sparse OTUs (99). Rarefaction (sequence depth of 11,535 sequences) was applied to the OTUs with QIIME to ensure the identical number of reads per sample in order to perform $\alpha$-diversity calculations using the Chao1 metric, Shannon index, and phylogenetic diversity (PD) (108–110).

**Statistical analyses.** The microbial diversity within each sample ($\alpha$-diversity) was assessed to investigate the overall microbiota development between P21, P42, and P180. The three-way generalized linear mixed model (GLMM) was performed at on the average Chao1, Shannon, and PD values at a sequencing depth of 11,535 sequences (highest possible sequencing depth at which $\alpha$-diversity could be calculated for all samples). Considering the low sample sizes of the P21 samples ($n$ = 3 to 5 per experimental group), these were excluded from further analyses. Therefore, a two-way GLMM was performed only at P42 and P180 separately, at a sequencing depth of 11,535 sequences.

The between-sample microbiota profile (dis)similarity ($\beta$-diversity) was assessed on the genus level, from aggregated OTU data, by permutational multivariate analysis of variance (PERMANOVA) using Bray-Curtis metrics, to assess age effects and clustering of the experimental groups. Additionally, distance-based redundancy analysis (db-RDA), also using Bray-Curtis metrics and performed on genus-level taxonomy, was used to assess the influence of conditions, diet, and their interactions on the fecal microbiota composition at each age separately. Since litter effects have been shown to drive gut microbiota variation in common laboratory mice (111), litter correction was applied to the db-RDA calculations. Data at the genus level was log-transformed and standardized by Hellinger transformation (112). Significance of the explained variance in the db-RDAs were assessed with an ANOVA-like permutation test for redundancy analysis (113). The 10 genera explaining the most variation in the db-RDA were visualized. The PERMANOVA and db-RDA procedures were performed using the vegan package (v.2.5-7) in R (v.3.6.2).

Next, the impact of conditions, diet, and age on the microbial taxa abundances was investigated. To this end, the sequence data were aggregated at the following taxonomic levels: genus, family, order, and phylum. Also, for microbial taxon abundances, litter was accounted for in the statistical analysis. GLMM was used to determine whether litter has a significant effect on the sequence data-derived abundances of a taxonomic group within every taxonomic level, and in cases where it did, it was taken along as a covariate in the GLMM. In order to estimate the effect of age, a three-way GLMM was performed on the sequence data-derived abundances of each taxonomic group within every taxonomic level, and in order to estimate the effect of conditions and diet, a two-way GLMM was employed on each age group (P42 and P180) separately. After performing a GLMM, the resulting sets of $P$ values, one set for each of the predictor variables and interactions thereof, were used to estimate the false-discovery rate (FDR) by calculating $q$ values (114). Resulting $P$ values of <0.05 with corresponding $q$ values of <0.1 were regarded as significant. Differences were visualized as bar plots with GraphPad Prism (v.9.1.2) and the Interactive Tree of Life (iTOL) v.3 (115).

**Correlational analysis between abundance of microbial species and central and peripheral outcomes.** Finally, using a Spearman correlation test, we tested whether the abundance of microbial species on four taxonomic levels from the current study correlated with previously reported parameters from the same mice (9). The parameters were cognitive behavior (performance on object location task and Morris water maze), fatty acid profiles in hippocampus, liver, and erythrocytes, and multiple metabolic outcomes (body weight, plasma leptin levels, inguinal fat, sum white fat). A detailed description of the methods regarding these parameters can be found in Yam et al. (9).

**Data availability.** The data that support the findings of this study are openly available in Figshare at https://doi.org/10.6084/m9.figshare.16748824.

## SUPPLEMENTAL MATERIAL

Supplemental material is available online only.
**FIG S1**, PDF file, 0.1 MB.
**FIG S2**, PDF file, 0.8 MB.
**TABLE S1**, PDF file, 0.04 MB.
**TABLE S2**, PDF file, 0.04 MB.
**TABLE S3**, PDF file, 0.1 MB.
**TABLE S4**, XLSX file, 0.1 MB.

## ACKNOWLEDGMENTS

We thank H. de Weerd for her contribution to the statistical analysis and P. van Leeuwen for his assistance with interpretation of the data.

S.T., M.M., J.K., and L.S. are employed by Danone Nutricia Research.

K.R. analyzed the data, prepared the figures, and wrote the manuscript. S.T. analyzed the data and prepared the figures. K.-Y.Y., L.S., and A.K. conceptualized the study, and K.-Y.Y. performed the mouse-related experimental work. M.M. contributed to correlation

analysis and discussion interpretation. A.K. supervised the study and reviewed and edited the manuscript. All authors contributed to editing of the manuscript.

This study was funded in part by Danone Nutricia Research.

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
