## [Reviewer comments · mSystems]

The role of the gut microbiota in the effects of early-life stress and dietary fatty acids on later-life central and metabolic outcomes in mice

Kitty Reemst, Sebastian Tims, Kit-yi Yam, Mona Mischke, Jan Knol, Stanley Brul, Lidewij Schipper, and Aniko Korosi

Corresponding Author(s): Aniko Korosi, Swammerdam Institute for Life Sciences

Review Timeline:

Submission Date:	February 19, 2022
Editorial Decision:	March 23, 2022
Revision Received:	April 25, 2022
Accepted:	April 25, 2022

Editor: Youjun Feng

Reviewer(s): Disclosure of reviewer identity is with reference to reviewer comments included in decision letter(s). The following individuals involved in review of your submission have agreed to reveal their identity: Beiwen Zheng (Reviewer #1)

Transaction Report:

DOI: <https://doi.org/10.1128/msystems.00180-22>

March 23, 2022

Dr. Aniko Korosi
Swammerdam Institute for Life Sciences
Center for Neuroscience
Amsterdam
Netherlands

Re: mSystems00180-22 (The role of the gut microbiota in the effects of early-life stress and dietary fatty acids on later-life central and metabolic outcomes in mice)

Dear Dr. Aniko Korosi:

Thank you for submitting your manuscript to mSystems. We have completed our review and I am pleased to inform you that, in principle, we expect to accept it for publication in mSystems. However, acceptance will not be final until you have adequately addressed the reviewer comments.

Current version is improved significantly, which have been approved by three of 4 reviewers. While, the data reliability and methodology is questioned by referee 4. Thus we offer you an opportunity to address the remaining comments and revise your manuscript accordingly!

Preparing Revision Guidelines

Sincerely,

Youjun Feng

Editor, mSystems

Journals Department
Reviewer comments:

Reviewer #2 (Comments for the Author):

The additions in the different sections improved the understanding of the study. I do not have any further comments.

Reviewer #3 (Comments for the Author):

The goal of this study was to identify whether polyunsaturated fatty acid (PFUA) ratio dietary changes could influence the microbiome of mice experiencing early life stress. The authors found that early life stress and low PFUA ratios can interact to modulate the beta diversity of the mouse gut microbiome.

The authors have performed an interesting and well-controlled study that builds nicely on previous work. I especially appreciate the experimental design of this study; the control groups used by the authors allow them to make important and interesting statistical comparisons. I especially appreciate the clarifications and additional explanations that the authors have added, particularly to the discussion section in terms of study limitations.

Comments

- I would recommend breaking figure 1 up into two different figures- one covering the study design and alpha diversity (current figure 1 panels A-D), and a second figure covering beta diversity (current figure 1 panels E-G). This should allow current figure 1 panels A-D to be larger and more visible (they are currently too small). It would also fit well into the manuscript structure, where alpha and beta diversity are discussed in different subsections of Results section 3.1.

Reviewer #4 (Comments for the Author):

SUMMARY: Reemst, et al. made substantive changes to Figure 1 and minor changes to Figures 2-4. However, despite the fact that it would not require any additional samples to be gathered, they did not perform the requested analyses to bring their bioinformatic analyses up to current standard practice. Because of this, all downstream analyses and conclusions likely do not represent the true community compositions, rendering all interpretations of the data unreliable. I do not recommend publication at this time.

I respectfully disagree with the authors about the use of OTU rather than sOTU/ASV. Removal of error from Illumina instruments is not new, and the practice has been field standard for years now (Callahan et al. ISME, 2017), (Amir et al. mSystems, 2017), (Callahan et al. Nat Methods, 2016), (Edgar, BioRxiv, 2017). In addition, please note that the authors of QIIME 1, on their website, state that "QIIME 2 has succeeded QIIME 1 as of January 1, 2018. QIIME 1 is no longer supported at this time" (<http://qiime.org>). In the absence of a valid reason why out of date methodologies are used for Illumina MiSeq data, trying to counteract their deleterious effects through filtering is not sufficient. According to independent evaluations and the original publications, denoising methods (DADA2, deblur) better resolve true community compositions. Therefore, the use of OTUs is no longer acceptable in the field. I highly recommend the authors consider reassessing the biological conclusions derived herein using modern methods and tools.

1. Nearing JT, Douglas GM, Comeau AM, Langille MGI. 2018. Denoising the Denoisers: an independent evaluation of microbiome sequence error-correction approaches. PeerJ 6:e5364 <https://doi.org/10.7717/peerj.5364>
2. Caruso V, Song X, Asquith M, Karstens L. Performance of Microbiome Sequence Inference Methods in Environments with Varying Biomass. mSystems. 2019 Feb 19;4(1):e00163-18. doi: 10.1128/mSystems.00163-18. PMID: 30801029; PMCID: PMC6381225.
3. Callahan BJ, McMurdie PJ, Holmes SP. Exact sequence variants should replace operational taxonomic units in marker-gene data analysis. The ISME Journal. 2017/12/01 2017;11(12):2639-2643. doi:10.1038/ismej.2017.119
4. Amir A, McDonald D, Navas-Molina JA, Kopylova E, Morton JT, Zech Xu Z, Kightley EP, Thompson LR, Hyde ER, Gonzalez A, Knight R. Deblur Rapidly Resolves Single-Nucleotide Community Sequence Patterns. mSystems. 2017 Mar 7;2(2):e00191-16. doi: 10.1128/mSystems.00191-16. PMID: 28289731; PMCID: PMC5340863.
5. Callahan BJ, McMurdie PJ, Rosen MJ, Han AW, Johnson AJ, Holmes SP. DADA2: High-resolution sample inference from Illumina amplicon data. Nat Methods. 2016 Jul;13(7):581-3. doi: 10.1038/nmeth.3869. Epub 2016 May 23. PMID: 27214047; PMCID: PMC4927377.

6. Edgar, R. C. (2017). UNOISE2: Improved error-correction for Illumina 16S and ITS amplicon sequencing. *bioRxiv*, 081257. <https://doi.org/10.1101/081257>

7. Knight R, Vrbanac A, Taylor BC, Aksenov A, Callewaert C, Debelius J, Gonzalez A, Kosciulek T, McCall LI, McDonald D, Melnik AV, Morton JT, Navas J, Quinn RA, Sanders JG, Swafford AD, Thompson LR, Tripathi A, Xu ZZ, Zaneveld JR, Zhu Q, Caporaso JG, Dorrestein PC. Best practices for analysing microbiomes. *Nat Rev Microbiol*. 2018 Jul;16(7):410-422. doi: 10.1038/s41579-018-0029-9. PMID: 29795328.

Here is a protocol on how to perform modern microbiome analysis using QIIME2:

• Estaki, M., Jiang, L., Bokulich, N. A., McDonald, D., González, A., Kosciulek, T., Martino, C., Zhu, Q., Birmingham, A., Vázquez-Baeza, Y., Dillon, M. R., Bolyen, E., Caporaso, J. G., & Knight, R. (2020). QIIME 2 enables comprehensive end-to-end analysis of diverse microbiome data and comparative studies with publicly available data. *Current Protocols in Bioinformatics*, 70, e100. doi: 10.1002/cpbi.100

A few example articles in your field that use updated methods after brief search on PubMed:

• Kemp KM, Colson J, Lorenz RG, Maynard CL, Pollock JS. Early life stress in mice alters gut microbiota independent of maternal microbiota inheritance. *Am J Physiol Regul Integr Comp Physiol*. 2021 May 1;320(5):R663-R674. doi: 10.1152/ajpregu.00072.2020. Epub 2021 Mar 3. PMID: 33655759; PMCID: PMC8163610.

• Differding MK, Doyon M, Bouchard L, Perron P, Guérin R, Asselin C, Massé E, Hivert MF, Mueller NT. Potential interaction between timing of infant complementary feeding and breastfeeding duration in determination of early childhood gut microbiota composition and BMI. *Pediatr Obes*. 2020 Aug;15(8):e12642. doi: 10.1111/ijpo.12642. Epub 2020 Apr 29. PMID: 32351036; PMCID: PMC7923600.

• Usui N, Matsuzaki H, Shimada S. Characterization of Early Life Stress-Affected Gut Microbiota. *Brain Sci*. 2021 Jul 10;11(7):913. doi: 10.3390/brainsci11070913. PMID: 34356147; PMCID: PMC8306161.

In addition, since the authors are working on updating methods, relative abundances are being phased out and log ratios are now much preferred in the field. Relative abundances are sum constraint-normalized data that increase probability of type I errors in analyses and make measurements of false discovery rates difficult. Relative abundance values can fluctuate significantly from study to study due to artifactual differences in the total number of microbial reads (i.e., total feature load) as well as method used (Nearing, et al. *Nat Commun*, 2022). For example, when the relative abundance of a specific bacterial family is increased, we cannot determine if this is due to an increase in the number of bacteria within that family or a decrease in the number of other bacteria in other families. By using log ratios for these analyses, we remove the biases created by the total feature load and can calculate false discovery rates using previously established methods. Thus, these log-ratio-based methods are more likely to result in repeatable trends in independently performed studies. Log ratios are calculated per sample on raw un-rarified read counts. To incorporate physiological data and microbiome compositional data, you should consider using compositional tensor factorization (CTF), DESeq2, ANCOM-BC, or ALDEx2. Other options include songbird (supervised) and deicode (unsupervised). These methods can be visualized and manipulated in qiime2/curro ("churro").

a. Morton JT, Marotz C, Washburne A, Silverman J, Zaramela LS, Edlund A, Zengler K, Knight R. Establishing microbial composition measurement standards with reference frames. *Nat Commun*. 2019 Jun 20;10(1):2719. doi: 10.1038/s41467-019-10656-5.

b. Martino C, Shenhav L, Marotz CA, Armstrong G, McDonald D, Vázquez-Baeza Y, Morton JT, Jiang L, Dominguez-Bello MG, Swafford AD, Halperin E, Knight R. Context-aware dimensionality reduction deconvolutes gut microbial community dynamics. *Nat Biotechnol*. 2021 Feb;39(2):165-168. doi: 10.1038/s41587-020-0660-7.

c. Fernandes, AD, Reid, JN, Macklaim, JM, McMurrugh, TA, Edgell, DR, & Gloor, GB. Unifying the analysis of high-throughput sequencing datasets: characterizing RNA-seq, 16S rRNA gene sequencing and selective growth experiments by compositional data analysis. *Microbiome*, 2, 15. (2014). <https://doi.org/10.1186/2049-2618-2-15>

d. Nearing JT, Douglas GM, Hayes MG, MacDonald J, Desai DK, Allward N, Jones CMA, Wright RJ, Dhanani AS, Comeau AM, Langille MGI. Microbiome differential abundance methods produce different results across 38 datasets. *Nat Commun*. 2022 Jan 17;13(1):342. doi: 10.1038/s41467-022-28034-z.

Example in your field:

• Coley EJL, Mayer EA, Osadchiy V, Chen Z, Subramanyam V, Zhang Y, Hsiao EY, Gao K, Bhatt R, Dong T, Vora P, Naliboff B, Jacobs JP, Gupta A. Early life adversity predicts brain-gut alterations associated with increased stress and mood. *Neurobiol Stress*. 2021 May 25;15:100348. doi: 10.1016/j.yjnstr.2021.100348. PMID: 34113697; PMCID: PMC8170500.

UNIVERSITY OF AMSTERDAM

*Swammerdam Institute for Life Sciences
Centre for Neuroscience
Brain plasticity group*

*Mail address: P.O.Box 94084
1090 GB Amsterdam*

*Visiting Address: Science Park 904
1098 XH Amsterdam, NL*

*Phone: (+31) 20-5257638 (secretariat)
Email: a.korosi@uva.nl*

to: Prof.Dr. Youjun Feng
Editor of *mSystems*

Amsterdam, April 25th 2022

Dear Prof. Dr. Youjun Feng,

Thank you for your letter informing us of the referees' positive impression of our paper; "The role of the gut microbiota in the effects of early-life stress and dietary fatty acids on later-life central and metabolic outcomes in mice" (mSystems 00180-22).

We are very pleased to hear that our work is in principle, acceptable for publication in mSystems, pending some minor modifications and we thank you and the reviewers for their time and input.

In your letter, you indicated that the current version has been approved by three of 4 reviewers, but that reviewer 4 asks for some further clarification on the data reliability and methodology. We address the points raised in detail below:

Reviewer #2; The additions in the different sections improved the understanding of the study. I do not have any further comments.

We thank the reviewer for the valued feedback and kind words.

Reviewer #3; The goal of this study was to identify whether polyunsaturated fatty acid (PFUA) ratio dietary changes could influence the microbiome of mice experiencing early life stress. The authors found that early life stress and low PFUA ratios can interact to modulate the beta diversity of the mouse gut microbiome.

The authors have performed an interesting and well-controlled study that builds nicely on previous work. I especially appreciate the experimental design of this study; the control groups used by the authors allows them to make important and interesting statistical comparisons. I especially appreciate the clarifications and additional explanations that the authors have added, particularly to the discussion section in terms of study limitations.

- I would recommend breaking figure 1 up into two different figures- one covering the study design and

alpha diversity (current figure 1 panels A-D), and a second figure covering beta diversity (current figure 1 panels E-G). This should allow current figure 1 panels A-D to be larger and more visible (they are currently too small). It would also fit well into the manuscript structure, where alpha and beta diversity are discussed in different subsections of Results section 3.1.

We thank the reviewer for the valued feedback and compliments. We agree about the size issue and have now increased the size of the bar plots and fonts in the Figure 1 panels A-D to increase readability. We decided not to split the entire figure, as it is deliberately set up to display the design and the alpha/beta diversity together in an integrated manner, and we feel that fragmenting it would not increase clarity.

Reviewer #4; SUMMARY: Reemst, et al. made substantive changes to Figure 1 and minor changes to Figures 2-4. However, despite the fact that it would not require any additional samples to be gathered, they did not perform the requested analyses to bring their bioinformatic analyses up to current standard practice. Because of this, all downstream analyses and conclusions likely do not represent the true community compositions, rendering all interpretations of the data unreliable. I do not recommend publication at this time.

We thank the reviewer for his/her comments. We would like to point out that in our first round of revision, we have implemented many, very detailed suggestions, which we would not qualify as ‘minor’ and which surely have greatly improved our manuscript. Regardless of this, referee 4’s major concern remains our analyses and use of qiime1, OTUs and relative abundances, versus the use of qiime2, aSVs and log ratios.

As mentioned in our first response letter while we are aware and agree on the importance of these new developments, we respectfully disagree with the reviewer that the existence of new tools makes the earlier developed, yet still valid and commonly used types of analysis, suddenly ‘unreliable’. We have explicitly explained our choices, and criteria under which our analysis was done, and discuss its advantages and limitations and are thus confident that our data and interpretations can be equally trusted and are of considerable value to the field in their current format.

In our revision, we discuss these developments, where the field of rRNA 16S amplicon sequencing has moved towards the use of sequence variants (aSVs) for data analysis (Glassman and Martiny 2018). As this especially impacts α -diversity measures (and OTUs tend to inflate this diversity (Glassman and Martiny 2018; Barnes et al 2020)), we have now further reduced any potential inflation of α -diversity measures, by filtering out all low abundant OTUs from our data (<0.002%)(see Auer et al 2016).

Secondly, and importantly, studies comparing OTU and aSV approaches have further demonstrated that when assessing concatenated data at a genus level, there is not much difference to be expected when genus level data is used originating either from OTUs or from aSVs (Glassman and Martiny 2018). Notably, as our analysis of the significance of taxa has been done at genus level or above, a new ‘aSV-based’ analysis is thus not expected to have a large effect on our results.

Concerning the aspect of the log-ratios vs the relative abundance: we believe that for our type of analysis (comparisons between experimental groups of mice where litter correction is important; and actually often neglected in similar studies), the use of relative abundance analyzed with a general linear model, in which we fit “litter” as a co-variate remains the most optimal statistical approach. In fact some of the tools proposed, like Aldex2 and ANCOM-BC, unfortunately do not have options to include the entire experimental design and correct for litter as well.

Thus, considering the above, we strongly feel that the efforts, time and resources it would cost to rewrite the entire paper following redoing all analyses -in full- using a new tool, as reviewer 4 suggests, does not outweigh the advantages of such re-analyses, will not change the final outcome much and will only further delay publication of the paper. We have taken the suggestions very seriously; we fully

acknowledge the importance of these analytical aspects by extensively and explicitly discussing also the limitations of our approaches, and mention future alternative options in our revised discussion (see below). Based on these arguments, we hope the referee and you, as mSystems editor will understand our current considerations.

“It is important to note that in our study we used the qiime v.1.9.0 pipeline⁴⁷ with which processing of sequencing data is performed based on OTUs. While commonly used, there are new developments in the field of rRNA 16S amplicon sequencing, where sequence variants (aSVs) are used for data analysis (Glassman et al., 2018). This is relevant as OTUs may inflate in particular measures of α -diversity (Barnes et al., 2020). In order to mitigate any such potential inflation, we have now filtered out low abundant OTUs (<0.002%)(Auer et al., 2017). Also, when assessing concatenated data at genus level, which we did in the current study, other studies indicate that there are no large differences to be expected when comparing genus level data originating either from OTU or from aSV data (Glassman et al., 2018). In fact, in this study, the analysis of phylogenetic β -diversity and the significance of taxa have all been studied at genus level or above. Taking this all together, we are confident about the reliability of our current analyses.”

Glassman, S. I. & Martiny, J. B. H. Broadscale Ecological Patterns Are Robust to Use of Exact Sequence Variants versus Operational Taxonomic Units. mSphere 3, (2018).

Barnes, C. J. et al. Comparing DADA2 and OTU clustering approaches in studying the bacterial communities of atopic dermatitis. J. Med. Microbiol. 69, 1293–1302 (2020).

Auer, L., Mariadassou, M., O’Donohue, M., Klopp, C. & Hernandez-Raquet, G. Analysis of large 16S rRNA Illumina data sets: Impact of singleton read filtering on microbial community description. Mol. Ecol. Resour. 17, e122–e132 (2017).

With that, we believe to have addressed all comments appropriately; we thank you again for the consideration of our manuscript and look forward to your response in due course.

On behalf of all authors, kind regards,

Kitty Reemst
Dr. Aniko Korosi

April 25, 2022

Dr. Aniko Korosi
Swammerdam Institute for Life Sciences
Center for Neuroscience
Amsterdam
Netherlands

Re: mSystems00180-22R1 (The role of the gut microbiota in the effects of early-life stress and dietary fatty acids on later-life central and metabolic outcomes in mice)

Dear Dr. Aniko Korosi:

Your manuscript has been accepted, and I am forwarding it to the ASM Journals Department for publication. For your reference, ASM Journals' address is given below. Before it can be scheduled for publication, your manuscript will be checked by the mSystems production staff to make sure that all elements meet the technical requirements for publication. They will contact you if anything needs to be revised before copyediting and production can begin. Otherwise, you will be notified when your proofs are ready to be viewed.

Publication Fees:

We recognize that the video files can become quite large, and so to avoid quality loss ASM suggests sending the video file via <https://www.wetransfer.com/>. When you have a final version of the video and the still ready to share, please send it to mSystems staff at mSystems@asmusa.org.

For mSystems research articles, if you would like to submit an image for consideration as the Featured Image for an issue, please contact mSystems staff at mSystems@asmusa.org.

Sincerely,

Youjun Feng
Editor, mSystems

Journals Department
Supplemental Table 2: Accept
Supplemental Figure 2: Accept
Supplemental Table 3: Accept
Supplemental Table S1: Accept
Supplemental Figure 1: Accept
Supplemental Table 4: Accept